CYP2C11 played a significant role in down-regulating rat blood pressure under the challenge of a high-salt diet

http://orcid.org/0000-0003-3362-1754 Liu Wei
Sui Danjuan
Ye Huanying
http://orcid.org/0000-0003-0246-9809 Ouyang Zhen
http://orcid.org/0000-0002-4025-6567 Wei Yuan ywei@ujs.edu.cn
School of Pharmacy, Jiangsu University , Zhenjiang , China
Khan Imran
Electronic publication date: 2019 Apr 23
Publication date: 2019
Volume: 7
Electronic Location ID: e6807
Received 2018 Dec 2; Accepted 2019 Mar 18
Copyright: © 2019 Liu et al.
Copyright year: 2019
Copyright holder: Liu et al.
License: This is an open access article distributed under the terms of the Creative Commons Attribution License, which permits unrestricted use, distribution, reproduction and adaptation in any medium and for any purpose provided that it is properly attributed. For attribution, the original author(s), title, publication source (PeerJ) and either DOI or URL of the article must be cited.
License URL: https://creativecommons.org/licenses/by/4.0/

Keywords: CYP2J2, Blood pressure, Epoxyeicosatrienoic acids, CYP2C11-null rats, High-salt diet

Funding: National Natural Science Foundation of China 81102522, 81373480, and 81573529 Natural Science Foundation of Jiangsu Province BK2011473 This work was funded by grants from the National Natural Science Foundation of China (Grant No. 81102522, 81373480, and 81573529) and the Natural Science Foundation of Jiangsu Province (Grant No. BK2011473). The funders had no role in study design, data collection and analysis, decision to publish, or preparation of the manuscript.

==============================
Background

Arachidonic acid (AA) is oxidized by cytochrome P450s (CYPs) to form epoxyeicosatrienoic acids (EETs), compounds that modulate ion transport, gene expression, and vasorelaxation. Both CYP2Cs and CYP2Js are involved in kidney EET epoxidation.

Methods

In this study, we used a CYP2C11-null rat model to explore the in vivo effects of CYP2C11 on vasorelaxation. For 2 months, CYP2C11-null and wild-type (WT) Sprague-Dawley rats were either fed normal lab (0.3% (w/w) sodium chloride) or high-salt (8% (w/w) sodium chloride) diets. Subsequently, an invasive method was used to determine blood pressure. Next, western blots, quantitative PCR, and immunohistochemistry were used to determine renal expression of CYPs involved in AA metabolism.

Results

Among CYP2C11-null rats, a high-salt diet (females: 156.79 ± 15.89 mm Hg, males: 130.25 ± 16.76 mm Hg, n = 10) resulted in significantly higher blood pressure than a normal diet (females: 118.05 ± 8.43 mm Hg, P < 0.01; males: 115.15 ± 11.45 mm Hg, P < 0.05, n = 10). Compared with WT rats under the high-salt diet, western blots showed that CYP2C11-null rats had higher renal expression of CYP2J2 and CYP4A. This was consistent with the results of immunohistochemistry and the qPCR, respectively. The two rat strains did not differ in the renal expression of CYP2C23 or CYP2C24.

Conclusion

Our findings suggested that CYP2C11 plays an important role in lowering blood pressure under the challenge of a high-salt diet.

Introduction

Hypertension is a common condition that can lead to ischemic heart disease, cerebrovascular disease, and chronic kidney disease (Collins et al., 1990; Van Gaal, Mertens & De Block, 2006; Wright, 2002). Therefore, regulating blood pressure is an effective way to reduce heart attack and heart failure incidence (Ambrosius et al., 2014; Peng et al., 2017; Zanchetti, Thomopoulos & Parati, 2015). Arachidonic acid (AA) is an omega-6 fatty acid with substantial effects on blood pressure in rats and humans. The kidneys primarily produce the following AA metabolites: regional and stereoisomeric epoxyeicosatrienoic acids (EETs) (Holla et al., 1999; Makita, Falck & Capdevila, 1996; Marji, Wang & Laniado-Schwartzman, 2002; Yu et al., 2000), as well as 20-hydroxyeicosatetraenoic acids (20-HETEs) (Imig, 2005; Marji, Wang & Laniado-Schwartzman, 2002; Zhao et al., 2003). AA CYP-epoxidase metabolites inhibit epithelial sodium channel activity in the cortical collecting duct (Capdevila et al., 2014; Capdevila & Wang, 2013), where 11,12-EET is primarily responsible for mediating the inhibition of epithelial Na channels. 11,12-EET also stimulates the Ca2+-activated large conductance potassium channel and afferent arteriolar smooth muscle in the cortical collecting duct (Capdevila & Wang, 2013).

In the kidneys, CYP4As (a type of cytochrome P450) catalyzes the formation of 20-HETE (Makita, Falck & Capdevila, 1996; Marji, Wang & Laniado-Schwartzman, 2002), which regulates vascular tone and electrolyte excretion (Imig, 2005). Through contracting peripheral blood vessels, 20-HETE promotes hypertension, but also inhibits Na and water absorption to exert anti-hypertensive effects (Marji, Wang & Laniado-Schwartzman, 2002). In rat kidneys, CYP2Cs and CYP2Js are the main producers of EETs (Zhao et al., 2003). The latter promotes microvascular dilation (Imig et al., 1996; Zhao et al., 2003), profibrinolytic effects, anti-inflammation, and inhibition of smooth muscle cell migration (Campbell, 2000; Fleming, 2001; Holla et al., 1999; Node et al., 2001). Among the cytochrome P450s, CYP2C11 epoxidizes AA to 11,12-EET and 14,15-EET with nearly equal efficiency, CYP2J mainly produces 14,15-EET, while CYP2C23 mainly epoxidizes AA to 11,12-EET, (Holla et al., 1999; Marji, Wang & Laniado-Schwartzman, 2002; Roman, 2002; Wójcikowski & Daniel, 2009; Yu et al., 2000). The latter is strongly influential in expanding blood vessels (Krötz et al., 2004; Node et al., 2001), suggesting that kidney CYP2Cs are important to controlling hypertension. The most highly expressed CYP2C in rat kidneys is CYP2C23, whereas CYP2C11 expression is much lower (Zhao et al., 2003). However, a portion of CYP2C23 is actually present in an inactivated form (Kaergel et al., 2002). Thus, the specific roles of CYP2Cs in EET-mediated blood-pressure regulation remain unclear.

Using a CYP2C11-null rat model that we had previously generated (Wei et al., 2018), here we examined how CYP2C11 influences EET function in vivo. We challenged CYP2C11-null and wild-type (WT) Sprague-Daly (S-D) rats with a high-salt diet to understand how blood pressure would change in response. We also investigated the expression of other P450s involved in renal AA metabolism, including CYP2C23, CYP2C24, CYP2J, and CYP4A.

Materials and Methods

Chemicals and reagents

Normal (0.3% (w/w) sodium chloride) and high-salt (8% (w/w) sodium chloride) rat grains were purchased from Xietong Organism (Nanjing, China). PageRuler™ Prestained Protein Ladder and RevertAid First Strand cDNA Synthesis Kit were purchased from Thermo Fisher Scientific (Waltham, MA, USA). Immobilon™ Western Chemiluminescent horseradish peroxidase (HRP) Substrate (ECL) was purchased from the EMD-Millipore (Billerica, MA, USA). CYP2J2 Rabbit Polyclonal antibody (Catalog number: 13562-1-AP) was purchased from Proteintech Group (Rosemont, IL, USA). The anti-CYP4A rabbit polyclonal antibody (catalog number: ab3573) was purchased from Abcam (Cambridge, UK). Sodium dodecyl sulfate-polyacrylamide gel electrophoresis (SDS-PAGE) sample loading buffer, enhanced bicinchoninic acid protein assay kit, and HRP-labelled goat anti-rabbit IgG (H+L) (catalog number: A0208) were purchased from Beyotime (Jiangsu, China). The HRP-polymer anti-mouse/rabbit antibody (catalog number: KIT-5030) and 3,3′-diaminobenzidine substrate were purchased from Maixin Biotech (Fuzhou, China). The UltraSYBR Mixture was purchased from ComWin Biotech (Beijing, China). DNase/RNase-free water was purchased from Solarbio Science & Technology (Beijing, China). Sodium phenobarbital (>98.5% purity) and all other reagents were purchased from Sinopharm Chemical Reagent (Shanghai, China).

Animals

The CYP2C11-null (knockout) rat model was generated using the CRISPR/Cas9 method to insert a two bp GT fragment into exon 6 of CYP2C11 (Wei et al., 2018). Rats were cultured at the Laboratory Animal Center of Jiangsu University (SPF grade). WT S-D rats (male and female, mean body weight 220 ± 10 g) were purchased from the Laboratory Animal Center of Jiangsu University (permit number SKXK (Su) 2013-0011). The breeding room environment was 22–25 °C with a relative humidity of 65% and a 12-h light/dark cycle. All subjects were allowed ad libitum access to water. Animal reproduction and experimental procedures were performed in accordance with the Guide for the Care and Use of Laboratory Animals and were approved by the Institutional Animal Care and Use Committee of Jiangsu University.

Animal experiments for measuring blood pressure

For the control (normal diet) condition, 2-month-old rats weighing 220 ± 10 g were divided into four groups, each containing 10 rats. Both sexes and strains (WT and CYP2C11-null) were represented equally. The same grouping method was used for the high-salt condition (Matrougui et al., 1998). Feeding duration was 2 months.

Blood pressure was measured using a classic invasive method (Schenk, Hebden & McNeill, 1992) with modifications. First, 150 mg/kg sodium phenobarbital was administered through an intraperitoneal injection. After disinfection, a small incision was made in the neck to reach the carotid artery. The artery’s upper part (near the brain) was tightened with a surgical line before inserting a thin tube with a three-way valve filled with heparin sodium solution (25 U/mL). The other end of the three-way valve was connected to a bio-signal collector. Blood pressure signals were collected using Chart version 5.0.1 (ADInstruments, NSW, Australia).

Western blots with rat kidney microsomes

Rats were euthanized with carbon dioxide after measuring their blood pressure. Renal microsomes were obtained with differential centrifugation (Ding & Coon, 1990). In brief, 0.5 g of kidney tissue were obtained and rinsed with ice-cold 0.9% sodium chloride solution. The tissue was then homogenized with six mL of KCl-sucrose buffer for 3 × 15 s and centrifuged at 9,000×g for 20 min at 4 °C. The resultant supernatant was then centrifuged at 100,000×g for 1 h at 4 °C. The resulting microsomal pellet was ground to a particle-free state after adding one mL of glycerol-phosphate buffer to a glass homogenizer and stored at −80 °C for use. The microsomal protein concentration was determined using a bicinchoninic acid kit (Beyotime, Jiangsu, China). Proteins (five µg) (Gu et al., 2003) were then subjected to 10% SDS-PAGE with duplicate adjacent lanes, and transferred to an Immobilon-P Transfer Membrane (EMD-Millipore, Burlington, MA, USA). Duplicate SDS-PAGE was performed and Ponceau S was used to ensure that sample proteins were loaded equally (Wang et al., 2017). The common reference proteins, β-actin and GAPDH, were not used because they are absent from renal microsomes. Membranes were blocked in 5% skim milk (Murraygoulburn Cooperative, Brunswick, VIC, Australia), probed with primary antibodies (1:1,000), and then incubated with species-specific HRP-conjugated secondary antibodies (Beyotime, 1:1,000). Signals were detected using an ECL system (Clinx Science Instruments, Shanghai, China). The western blotting analysis was repeated three times and images were analyzed with Image J software (version 1.46, National Institutes of Health, Bethesda, MD, USA).

Quantitative polymerase chain reaction analysis

Because CYP2C23 and CYP2C24 antibodies were unavailable, renal mRNAs were analyzed to determine expression of these CYPs. Renal mRNA levels of CYP2J2 and CYP4A1 were also analyzed. Total RNA was isolated using a TRIzol reagent (ComWin Biotech, Beijing, China) following the manufacturer’s protocol. Complementary DNA was synthesized from five μg of RNA using a cDNA Synthesis Kit (Thermo Fisher Scientific, Waltham, MA, USA) and eluted in a final volume of 20 μL. Quantitative polymerase chain reaction (qPCR) was performed using the LightCycler96 Real-time PCR system (Roche, Basel, Switzerland). Primers (Table 1) for CYP2C23, CYP2C24, and β-actin were synthesized by Sangon Biotech (Shanghai, China). The thermocycling profile was 95 °C for 600 s, followed by 46 cycles of 95 °C for 15 s, 61 °C for 20 s, and 72 °C for 30 s. The 2−ΔΔCt method was used to calculate the relative changes in target gene expression from the qPCR results, using β-actin as the reference gene.

Table 1 Sequences of the real-time PCR primers.

Gene	Primer sequence	
CYP2C23	5′-GGCTGTCTGTGGGTCTAACT-3′	
5′-AGGCAGCTTCATCTTGTCCT-3′	
CYP2C24	5′-AGGCTGTCATGGATCCAGTC-3′	
5′-AGGCTTCGGACCAAAGTACA-3′	
CYP2J2	5′-TTCCTACTCCTGGCTGAC-3′	
5′-TTTATTGGTGATGCGTTC-3′	
CYP4A1	5′-GCAGTGTTCAGGTGGAT-3′	
5′-CTGGCAAGTAAGAGGATG-3′	
β-actin	5′-TGGAATCCTGTGGCATCCATGAAAC-3′	
5′-TAAAACGCAGCTCAGTAACAGTCCG-3′	

Immunohistochemistry

Fixed kidney sections were de-waxed and rehydrated following published methods (Becker et al., 2006). Endogenous peroxidase was inactivated through 0.3% hydrogen peroxide treatment for 10 min. Treatment with phosphate-buffered saline for 5 min blocked non-specific binding. Sections were then incubated with primary antibodies (CYP2J2, 1:100; CYP4A, 1:300) overnight at 4 °C, followed by the secondary antibody (Maixin, 1:20) for 20 min. Sections were stained for 2 min with 3,3′-diaminobenzidine substrate, then rinsed with distilled water before being dehydrated and covered in a neutral resin/xylene mixture. Section images were obtained under 200× magnification in five randomly selected microscopic fields and analyzed in Image-Pro Plus version 6.0 (Media Cybernetics, Rockville, MD, USA). The integral optical density (IOD) and total area of the positive reaction region in each field of vision were automatically generated by the software. Their ratio was the mean density (IOD/area).

Statistical analysis

All data are expressed as means ± standard deviation. Student’s t-test was used to determine differences in means. Analyses were performed in Prism 6.0 (GraphPad, La Jolla, CA, USA), and P < 0.05 was considered statistically significant.

Results

High-salt diets elevated blood pressure in CYP2C11-null rats

Under a normal lab diet, CYP2C11-null and WT rats did not differ in blood pressure (n = 10). Among CYP2C11-null rats, male and female average blood pressure was 115.2 ± 11.5 mm Hg and 118.1 ± 8.4 mm Hg, respectively. Among WT, male and female blood pressure was 120.4 ± 11.4 mm Hg and 120.0 ± 16.5 mm Hg. However, a high-salt diet significantly increased blood pressure (male: 130.3 ± 16.8 mm Hg, female: 156.8 ± 15.9 mm Hg; n = 10) among CYP2C11-null rats compared with a normal diet (male: 115.2 ± 11.5 mm Hg, P < 0.05, female: 118.1 ± 8.4 mm Hg, P < 0.01; n = 10). The increase in CYP2C11-null females was significantly higher than in CYP2C11-null males and WT females (128.5 ± 21.8 mm Hg, n = 10, P < 0.01) (Fig. 1).

Figure 1 High-salt diets elevated blood pressure in CYP2C11-null rats.

A high-salt diet significantly increased blood pressure among CYP2C11-null rats compared with a normal diet (female, P < 0.01, male, P < 0.05; n = 10). The increase in CYP2C11-null females was significantly higher than in CYP2C11-null males and WT females (P < 0.01, n = 10). All data are expressed as mean ± standard deviation. *P < 0.05, **P < 0.01.

Western blots of CYP2J2 and CYP4A in rat kidney microsomes

Under the normal diet, the expression of CYP2J2 in CYP2C11-null male rats was significantly higher than that in WT male rats. In addition, the expression of CYP4A in CYP2C11-null female rats was significantly lower than that in WT female rats (P < 0.01). The high-salt diet significantly increased renal CYP2J2 and CYP4A expression in both male and female CYP2C11-null rats compared with WT rats (P < 0.01) (Fig. 2).

Figure 2 Western blots of CYP2J2 and CYP4A in rat kidney microsomes.

Representative western blots showing 57-kDa CYP2J2 and 50-kDa CYP4A protein bands in CYP2C11-null rats and WT rats in the normal (A and B) and high-salt (C and D) diet. Proteins were loaded in duplicate adjacent lanes. Under the normal diet, the expression of CYP2J2 (E) in CYP2C11-null male rats was significantly higher than that in WT male rats. In addition, the expression of CYP4A (G) in CYP2C11-null female rats was significantly lower than that in WT female rats (P < 0.01). The high-salt diet significantly increased renal CYP2J2 (F) and CYP4A (H) expression in both male and female CYP2C11-null rats compared with WT ones. Integrated density of the WT rats was regarded as 100%. All data were expressed as means ± standard deviation. **P < 0.01.

Renal mRNA expression

After knocking out the CYP2C11 gene, renal CYP2C23, and CYP2C24 mRNA did not differ between CYP2C11-null and WT rats of either sex (n = 9). The results showed that the high-salt diet treatment had no effects on renal CYP2C23 or CYP2C24 mRNA expression. The mRNA levels of these genes in the two rat strains under the normal diet were consistent with the results of western blotting. Under the high-salt diet, renal expression of CYP2J2 and CYP4A1 mRNA in CYP2C11-null male rats was significantly higher than that in WT male rats (P < 0.01, n = 9) (Fig. 3).

Figure 3 Quantitative PCR of renal mRNA expression.

After knocking out the CYP2C11 gene, renal CYP2C23 (A and B), and CYP2C24 (C and D) mRNA did not differ between CYP2C11-null and WT rats of either sex (n = 9). The results showed that the high-salt diet treatment had no effects on renal CYP2C23 or CYP2C24 mRNA expression. Under the normal diet, the mRNA expression of CYP2J2 (E) in CYP2C11-null male rats was significantly higher than that in WT male rats (P < 0.01, n = 9), and the mRNA expression of CYP4A1 (G) in CYP2C11-null female rats was significantly lower than that in WT female rats (P < 0.01, n = 9). Under the high-salt diet, renal expression of CYP2J2 (F) and CYP4A1 (H) mRNA in CYP2C11-null male rats was significantly higher than that in WT male rats (P < 0.01, n = 9). All data were expressed as means ± standard deviation. **P < 0.01.

Immunohistochemistry

Positive staining for CYP2J2 was observed in the renal cortex and outer medulla of all groups. The positive reaction of CYP2C11-null female rats treated with the normal diet was enhanced compared to WT female rats. The positive reaction and staining intensity of the CYP2C11-null rats treated with a high-salt diet were increased compared to WT rats. Renal sections of all groups were observed to be positively stained for CYP4A in interlobar, arcuate, and interlobular arteries. However, no positive enhancement was observed in the sections (Fig. 4A). Under a normal diet, renal CYP2J2 expression was significantly elevated in CYP2C11-null females compared with WT females (P < 0.01, n = 5). We observed the same between-strain difference under the high-salt diet, except in both sexes (P < 0.01, n = 5). No significant difference was found for CYP4A expression (Fig. 4B).

Figure 4 Immunohistochemistry (cytoplasmic, positive) of CYP2J2 and CYP4A (magnification, ×200).

(A) Positive staining for CYP2J2 (A–H) was observed in the renal cortex and outer medulla of all groups. Under the normal diet, the positive reaction of D group was enhanced compared to B group. Under the high-salt diet, the positive reaction and staining intensity of G and H groups were increased compared to E and F groups. In tissue sections with arrow marks, the arrow a indicated the cortical staining area and the arrow b indicated the outer medulla stained area. Renal sections of all groups were observed to be positively stained for CYP4A (I–P) in interlobar, arcuate, and interlobular arteries. However, no positive enhancement was observed in the sections. (B) Under the normal diet, renal CYP2J2 expression (Q) was significantly elevated in CYP2C11-null females compared with WT females (P < 0.01, n = 5). We observed the same between-strain difference under the high-salt diet, except in both sexes (P < 0.01, n = 5). No significant difference was found for CYP4A expression (R). Density (mean) = integrated optical density (IOD)/Area. All data were expressed as means ± standard deviation. **P < 0.01.

Discussion

In this study, we provided the first in vivo evidence of CYP2C11 function in EET-mediated blood pressure regulation. Our results showed that high-salt diets only elevated blood pressure in CYP2C11 knockout rats, but did not affect blood pressure in WT S-D rats. Thus, high sodium alone was insufficient for blood-pressure increases, consistent with previous reports (Zhao et al., 2003), and rats without CYP2C11 were more vulnerable to the challenge of the high-salt diet.

In the kidney, CYPs generate two major classes of AA metabolites: HETEs (20- and 19-HETE) (Fleming, 2001) and EETs (5,6-, 8,9-, 11,12-, and 14,15-EET) (Capdevila, Falck & Harris, 2000; Capdevila et al., 1992; Makita, Falck & Capdevila, 1996; Wu et al., 1997). In particular, CYP4A generates 20-HETE, a ω-hydroxylation product of AA that is important to blood pressure regulation (Müller et al., 1996; Roman, 2002). Previous research has provided clear evidence of CYP4A involvement in 20-HETE production (Escalante, McGiff & Oyekan, 2002; Oyekan et al., 1999; Wang et al., 2003), as the inhibition of nitric oxide synthase restored elevated blood pressure. Removing nitric oxide increases ω-hydroxylase activity (Wang et al., 2003), CYP4A protein expression (Oyekan et al., 1999), and 20-HETE production in the kidney (Oyekan et al., 1999; Wang et al., 2003). In the present study, western blots showed that the high-salt diet significantly elevated renal CYP2J2 and CYP4A expression, but immunohistochemistry only revealed a significant between-group difference in renal CYP2J2 expression under the high-salt diet. Because the high-salt diet significantly elevated renal CYP4A mRNA expression, the salt-induced elevation in blood pressure might be slightly compensated by the formation of 20-HETE.

We also demonstrated that CYP2C11 metabolites had important physiological functions. Although equipment and technical limitations prevented us from monitoring EETs and 20-HETEs, we found that CYP2C11 metabolites had important physiological functions in the regulation of blood pressure in rats. EETs are known contributors to kidney function through their direct effects on tubular transport processes, vascular tone, and cellular proliferation (Fisslthaler et al., 2000; Imig et al., 1996). The ratio of vasodilating (primarily 11,12- and 14,15-EET) to vasoconstrictive EETs (primarily 5,6-EET) controlled which process occurred (Yu et al., 2000). Of these EETs, 5,6 metabolites are extremely unstable, causing difficulties for measuring 5,6-EET formation in vitro and in vivo (Yu et al., 2000). In rat renal microsomes, only a negligible amount of 5,6-EET was found due to low production from CYP2C11, CYP2C23, CYP2C24, and CYP2J (Capdevila et al., 1992; Holla et al., 1999; Karara et al., 1993; Wu et al., 1997). The 8,9-, 11,12-, and 14,15-EETs were considered to have important physiological functions. CYP2C11 catalyzed the oxidation of 11, 12-, and 14, 15-EETs with almost the equal efficiency (Holla et al., 1999). However, the highly expressed CYP2C23 catalyzed the formation of 8,9-, 11,12-, and 14,15-EET at a ratio of approximately 1:2:1 in the rat kidney (Karara et al., 1993). Furthermore, CYP2J isoforms preferentially catalyzed AA to 14,15-EET (Zhao et al., 2003).

In our study, CYP2C11 knockout did not affect renal CYP2C23 and CYP2C24 mRNA expression. Thus, these two proteins are highly unlikely to be involved in the processes that elevate blood pressure after a high-salt diet. Similarly, research comparing spontaneously hypertensive rats to WKY rats shows that CYP2C11, CYP2C23, and CYP2C24 are expressed at similar levels in the renal cortex (Yu et al., 2000). Furthermore, a high-salt diet induced CYP2J expression in CYP2C11-null rats but not in WT rats, consistent with the finding that CYP2J is elevated in spontaneously hypertensive rats (Yu et al., 2000). Therefore, inducing CYP2J2 expression in CYP2C11 knockout rats caused a slight compensatory effect that was nevertheless insufficient to fully replace CYP2C11.

Rat CYP2C11 is expressed specifically in males and barely present in females. Thus, knockout of CYP2C11 might cause greater compensatory effects from other P450s in males. We speculate that, under the challenge of a high-salt diet, the compensatory effects in CYP2C11-null females were exhausted, resulting in a more significant increase in blood pressure compared to that in CYP2C11-null males. Compared with WT females, CYP2C11-null females also had a more significant increase in blood pressure under the high-salt diet. This could be explained by the function of CYP2C11 in WT females, despite its very low expression level.

Conclusions

CYP2C11 plays a key role in mitigating the elevated blood pressure resulting from a high-salt diet. We confirmed that in vivo CYP function is associated with sodium-induced hypertension. Although more research is necessary to confirm the relevance of CYP to sodium-induced hypertension in humans, our results corroborate human research showing that CYP2J2 (Wu et al., 2007) and CYP4A11 (Fu et al., 2008) are related to hypertension. Additionally, our CYP2C11-null rat model should benefit further study on this topic.

Supplemental Information

Supplemental Information 1 Raw Data for rat blood pressure, western blots, qPCR, and immunohistochemistry.

Includes the body weight data at Day 0 and the day of measuring blood pressure.

Click here for additional data file.

Supplemental Information 2 Full length uncropped western blots.

All uncropped and modified figures of rat western blots on a high-salt diet and a normal diet.

Click here for additional data file.

We would like to thank Professor Miao Chen of the Department of Pathology, First People’s Hospital, Zhenjiang, China, for immunohistochemistry analysis of rat tissue samples.

Additional Information and Declarations

Competing Interests

Author Contributions

Animal Ethics

Data Availability

The authors declare that they have no competing interests.

Wei Liu conceived and designed the experiments, performed the experiments, analyzed the data, contributed reagents/materials/analysis tools, prepared figures and/or tables, authored or reviewed drafts of the paper, approved the final draft.

Danjuan Sui performed the experiments, contributed reagents/materials/analysis tools, approved the final draft.

Huanying Ye performed the experiments, contributed reagents/materials/analysis tools, approved the final draft.

Zhen Ouyang authored or reviewed drafts of the paper, approved the final draft.

Yuan Wei conceived and designed the experiments, analyzed the data, prepared figures and/or tables, authored or reviewed drafts of the paper, approved the final draft.

The following information was supplied relating to ethical approvals (i.e., approving body and any reference numbers):

Animal reproduction and experimental procedures were performed in accordance with the Guide for the Care and Use of Laboratory Animals and were approved by the Institutional Animal Care and Use Committee of Jiangsu University.

The following information was supplied regarding data availability:

The raw measurements are available in File S1.

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
