# Peer review of "CYP2C11 played a significant role in down-regulating rat blood pressure under the challenge of a high-salt diet"

_PeerJ, doi:10.7717/peerj.6807_

## Round 0.1 · original submission · Major Revisions

Both the reviewers have raised several question. Please revise the manuscript incorporating all the requested changes.

Reviewer 1 ·

Basic reporting

The manuscript entitled "CYP2C11 played a significant role in down regulating rat blood pressure under the challenge of a high-salt diet" by Wei Liu et al. is a good observational manuscript. The rat model for CYP is crucial to decipher the salt diet and blood pressure relation. However, the mechanistic insight is lacking, which may be acceptable since it is first correlational study. Language of the manuscript is clear and satisfactory with very few grammatical errors. I do recommend it for publication only if authors address the following concern list below.

Experimental design

I have few concerns with experimental design, which I had listed below. qPCR should have been performed for all four genes. Explanation of author for not performing qPCR for two other genes is not acceptable.

Validity of the findings

Findings may have a broader impact.

Additional comments

Line 29: Our not our
Line 30: remove ‘a’

In result section: Authors are advised to compare each subject sequentially throughout the manuscript. For eg. If you are comparing male and female data as in line 150 and 151, keep the sequence same in line 153 and 154 and not jumble it like female and male.
In feeding duration of 2 months, did authors have checked the change in weight pattern of mice? Also, what was the weight of mice at the time/day of measuring blood pressure?

In the western blot result:
1. There is no loading control in figure 2, although author explained that common control proteins like β-actin and GAPDH are absent from renal microsomes, Ponceau staining of blot can be used instead for the same.

2. Without quantification of western blot signal, it is difficult to conclude anything from the data. Quantification of blot signal is required for comparison and can be done by freely available NIH ImageJ software.

3. Also mention the molecular weights in figure 2 themselves, even it is mentioned in legend. Describe each lane individually since it is unclear whether the samples were loaded in duplicate in adjacent lanes or samples were from two individual?

4. Please explain in detail the microsome isolation in the methods section, although reference was cited for the same.

5. Did author have used any quality control for the purity of microsomal fractions? Since, β-actin and GAPDH are not supposed to be present in microsomal fractions, western blot with these antibodies could serve as an excellent control for the purity of microsomes.

Line 163: CTP2C11 or CYP2C11?

In qPCR:
Author chose to quantify the renal mRNA expression level of CYP2C23 and CYP2C24 genes only, since antibodies of these two proteins were not present. It would be more comprehensive if mRNA expression level of all the four genes i.e, CYP2C23, CYP2C24, CYP2J, and CYP4A were analyzed and western blot of CYP2J and CYP4A can complement the findings.

Immunohistochemistry section needs more attention. Author should explain their result and observation in more detail. Authors have labeled each figure but didn’t explain any one of them in result section. For which parameters, images were analyzed by image-pro plus version 6, explain them in detail in result section. Also, convert table 2 in bar diagram and merge with figure 4.

Reviewer 2 ·

Basic reporting

Liu el al, investigated the in vivo effect of CYP2C11 function in epoxyeicosatrienoic acids (EET)-mediated blood pressure regulation and showed that high-salt diets only elevate rat blood pressure in CYP2C11 knockout rats. Authors conclude that high salt alone is insufficient for blood-pressure increase and rats without CYP2C11 are more vulnerable to the challenge of high-salt diet. Authors investigated the expression of various P450s involved in renal Arachidonic acid (AA) metabolism, including CYP2C23, CYP2C24, CYP2J, and CYP4A. Overall authors try to understand the important of cytochrome P450s in regulating blood samples. Manuscript is written clearly with sufficient background, however I have concern for the conclusions drawn from the current data.

Experimental design

Methods are described with sufficient detail and information.

Validity of the findings

Authors should add sufficient detail in figure legends and also add information about the controls used in each experiment.

Additional comments

Comments :
(1). "Under a normal lab diet, CYP2C11-null and WT rats did not differ in blood pressure. However, a high-salt diet significantly increased blood pressure among CYP2C11-null rats compared with a normal diet. The increase in CYP2C11-null females was significantly higher than in CYP2C11-null males and WT females". Can authors describe/speculate the reason for this variability ?

(2). Figure 2, under high-salt diet, renal CYP2J2 expression increased both in male and female CYP2C11-null rats compared with WT rats. Authors suggest no significant effect on CYP4A renal expression, however in CYP4A lanes, I can see two very undifferentiated bands both in male and female samples and these bands intensity increased under high-salt condition.
My concerns are following :
a) Authors should show loading control for each sample
(b) Quantify the data by normalizing to loading control
(c) Gel should be run longer to see whether there is single or two bands for CYP4A. These could be two isoforms of CYP4A. After quantifying data, authors can reach to conclusion that there is no change in CYP4A level under high salt condition.
(d) What about the protein expression for CYP2C23 and CYP2C24 ?

(3). In figure 3, authors suggest that after knocking out the CTP2C11 gene, renal CYP2C23 and CYP2C24 mRNA did not differ between CYP2C11-null and WT rats and diet had no effect on mRNA expression. However, I observed that authors have different scales for y-axis in each sample, hence it is difficult to compare the expression level by looking at current graphs. By looking at data in current graphs, I see that expression level of CYP2C23 is lower under high salt diet whereas expression level of CYP2C24 is high in both null mutants. To make right conclusion about the expression level and significance change, authors should have similar Y-axes and show the mean value/or other parameter for each sample. How the mRNA levels correlate for CYP4A and CYP2J2?

(4). It is difficult to follow the variability in figure 4. Authors should add arrows to indicate variability and also add description in the figure legend.

---

## Round 0.2 · accepted · Accept

Authors have addressed all the questions raised by reviewers. Manuscript can be accepted for publication.

# Reviewer 1 ·

Basic reporting

The manuscript has improved significantly, but still need few improvement:
Result:
Results subheading should be conclusive as in line 160, so make changes accordingly in all further result subheadings.

Experimental design

Not applicable

Validity of the findings

Already provided.

Additional comments

The explanation for the comment mentioned below is still not substantial to consider for the purity of microsome fractions. Both the articles recommended by author doesn't provide the protocol for microsomes isolation and their purity and above that referred paper cite the other paper for the protocol. I would suggest the author to refer the correct paper and methodology and proper controls and accordingly mention them all in method section.

5. Did author have used any quality control for the purity of microsomal fractions? Since, β-actin and GAPDH are not supposed to be present in microsomal fractions, western blot with these antibodies could serve as an excellent control for the purity of microsomes.
We didn’t use any quality control for the purity of the microsomes. This method was learned from my Ph.D. advisor Dr. Xinxin Ding (Department of Pharmacology and Toxicology, University of Arizona) and had been used for many years. This method worked well in our previous studies. (For example: Wei et al., 2010; Wei et al., 2018; etc.)

Reviewer 2 ·

Basic reporting

Manuscript is clearly written.

Experimental design

In revised manuscript, authors have added relevant detail in method section

Validity of the findings

Data is robust and conclusions are well stated.

Additional comments

Relevant changes have been included in revised manuscript.